Triassic pentadactyl tracks from the Los Menucos Group (Río Negro province, Patagonia Argentina): possible constraints on the autopodial posture of Gondwanan trackmakers

Citton Paolo pcitton@unrn.edu.ar 1 2
Díaz-Martínez Ignacio 1 2
de Valais Silvina 1 2
Cónsole-Gonella Carlos 1 3
1 Consejo Nacional de Investigaciones Científicas y Técnicas (CONICET) , Buenos Aires , Argentina
2 Instituto de Investigación en Paleobiología y Geología (IIPG), Universidad Nacional de Río Negro , General Roca , Argentina
3 Instituto Superior de Correlación Geológica (INSUGEO), Universidad Nacional de Tucumán , Tucumán , Argentina
De Baets Kenneth
Electronic publication date: 2018 Aug 7
Publication date: 2018
Volume: 6
Electronic Location ID: e5358
Received 2018 Mar 19; Accepted 2018 Jul 11
Copyright: ©2018 Citton et al.
Copyright year: 2018
Copyright holder: Citton et al.
License: This is an open access article distributed under the terms of the Creative Commons Attribution License, which permits unrestricted use, distribution, reproduction and adaptation in any medium and for any purpose provided that it is properly attributed. For attribution, the original author(s), title, publication source (PeerJ) and either DOI or URL of the article must be cited.
License URL: https://creativecommons.org/licenses/by/4.0/

Keywords: Pentasauropus, Tracks, Therapsids, Dicynodonts, Triassic, Gondwana, Los Menucos, Patagonia, Trackmakers

Funding: Paleontological Society International Research Program CONICET to Silvina de Valais PIP 2015-2017 576 This work was made possible by financial support from the Paleontological Society International Research Program, Sepkoski Grants –2017 to Paolo Citton and PIP 2015-2017 576 from CONICET to Silvina de Valais. The funders had no role in study design, data collection and analysis, decision to publish, or preparation of the manuscript.

==============================
The Los Menucos locality in Patagonia, Argentina, bears a well-known ichnofauna mostly documented by small therapsid footprints. Within this ichnofauna, large pentadactyl footprints are also represented but to date were relatively underinvestigated. These footprints are here analyzed and discussed based on palaeobiological indications (i.e., trackmaker identification). High resolution digital photogrammetry method was performed to achieve a more objective representation of footprint three-dimensional morphologies. The footprints under study are compared with Pentasauropus from the Upper Triassic lower Elliot Formation (Stormberg Group) of the Karoo Basin (Lesotho, southern Africa). Some track features suggest a therapsid-grade synapsid as the potential trackmaker, to be sought among anomodont dicynodonts (probably Kannemeyeriiformes). While the interpretation of limb posture in the producer of Pentasauropus tracks from the Los Menucos locality agrees with those described from the dicynodont body fossil record, the autopodial posture does not completely agree. The relative distance between the impression of the digital (ungual) bases and the distal edge of the pad trace characterizing the studied tracks likely indicates a subunguligrade foot posture (i.e., standing on the last and penultimate phalanges) in static stance, but plantiportal (i.e., the whole foot skeleton and related soft tissues are weight-bearing) during the dynamics of locomotion. The reconstructed posture might have implied an arched configuration of the articulated metapodials and at least of the proximal phalanges, as well as little movement capabilities of the metapodials. Usually, a subunguligrade-plantiportal autopod has been described for gigantic animals (over six hundreds kilograms of body weight) to obtain an efficient management of body weight. Nevertheless, this kind of autopod is described here for large but not gigantic animals, as the putative trackmakers of Pentasauropus were. This attribution implies that such an autopodial structure was promoted independently from the body size in the putative trackmakers. From an evolutionary point of view, subunguligrade-plantiportal autopods not necessarily must be related with an increase in body size, but rather the increase in body size requires a subunguligrade or unguligrade, plantiportal foot. Chronostratigraphically, Pentasauropus was reported from Upper Triassic deposits of South Africa and United States, and from late Middle Triassic and Upper Triassic deposits of Argentina. Based on the stratigraphic distribution of the ichnogenus currently accepted, a Late Triassic age is here proposed for the Pentasauropus-bearing levels of the Los Menucos Group.

Introduction

Tetrapod tracks are valuable fossils which inform us about the anatomy (e.g., Carpenter, 1992), functional adaptations (e.g., Baird, 1980), motion (e.g., Avanzini, Piñuela & García-Ramos, 2011) and ethology (e.g., Lockley et al., 2016) of extinct animals, greatly expanding the potential of information that is often precluded from the body-fossil record. The detailed analysis of tetrapod footprints is therefore significant for integrating and revising data derived from the tetrapod body-fossil record.

The scientific study of tetrapod footprints in Argentina is relatively recent compared to that of Europe (Duncan, 1831; Kaup, 1835a; Kaup, 1835b) and North America (Hitchcock, 1836), dating back to the first half of the twentieth century (Von Huene, 1931). One of the most important contribution to tetrapod ichnology in Argentina is that of Casamiquela (1964), who devoted himself to the study of Triassic and Jurassic tetrapod tracks from Patagonia. Later, other contributions focused on important Triassic ichnofaunas from other regions of Argentina have been published (e.g., Romer, 1966; Bonaparte, 1966; Leonardi, 1994; Melchor & de Valais, 2006). Among the Triassic vertebrate ichnological record, the Los Menucos ichnofauna, which is dominated by small therapsid footprints, was repeatedly studied (Casamiquela, 1964; Casamiquela, 1975; Casamiquela, 1987; Leonardi & De Oliveira, 1990; Leonardi, 1994; Domnanovich & Marsicano, 2006; Melchor & de Valais, 2006; de Valais, 2008; Domnanovich et al., 2008; Díaz-Martínez & de Valais, 2014). The bulk of this ichnofauna was originally attributed to different ichnotaxa by Casamiquela (1964) and Casamiquela (1975), but after the revision made by Melchor & de Valais (2006), most of the ichnogenera erected by Casamiquela are considered synonymous with Dicynodontipus. Moreover, an indetermined chirotheroid track (de Valais, 2008), a single track referred to as Rhynchosauroides, and large pentadactyl footprints mentioned as Pentasauropus sp. (Domnanovich et al., 2008) have also been reported from the Los Menucos area. From the same locality, several slabs with pentadactyl tracks comparable to those described by Domnanovich et al. (2008) were collected many years ago but remained unpublished until now.

An ichnological analysis based on this material is here proposed and discussed in terms of the palaeobiology, identity and autopodial anatomy of the trackmaker. Besides, a brief discussion of the chronostratigraphy of this record is provided.

Material and Methods

The present study is based on the direct examination of track-bearing slabs MPCA 27029-1 with three pes-manus couples, two of which incomplete (concave epireliefs, i.e., negative relief), MPCA 27029-2 with a single left pes-manus couple (convex hyporeliefs, i.e., positive relief), MPCA 27029-3 with two pes-manus couples and an incomplete pes (convex hyporeliefs), MPCA 27029-4 with a single pes-manus couple (convex hyporeliefs), MPCA 27029-5 with a single track (convex hyporeliefs), MPCA 27029-9 with five pes-manus couples, three of which incomplete (convex hyporeliefs), MPCA 27029-16 with three pes-manus couples, one of which incomplete, and five incomplete tracks (convex hyporeliefs), MPCA 27029-21 with two pes-manus couples and four tracks (convex hyporeliefs), MPCA 27029-33 with two pes-manus couples (convex hyporeliefs), MMLM 1 with two pes-manus couples (convex hyporeliefs), MMLM 2 with two incomplete pes-manus couples (convex hyporeliefs), and MMLM 075-1 (ex MRPV 1987P.V.06 in Domnanovich et al., 2008, hereafter MMLM 075-1) with two incomplete pes-manus couples (concave epireliefs). Except for the specimen MMLM 075-1, the material under study was to date unpublished. A few other slabs, both with and without label, are stored at the MPCA but were not considered in this study due to poor preservation of the tracks. In total, about 60 footprints were analyzed. For each slab, tracks were numbered using Arabic numerals and, when referring to single tracks in the text, they are indicated as /number following the slab label (e.g., MPCA 27029-1/4 where MPCA 27029-1 and number 4 indicate slab and single track, respectively). The studied material mainly consists of isolated sets or incomplete trackways.

The provenance of the track-bearing slabs can be traced back to the Felipe Curuil ex quarry, Yancaqueo farm, east of the town of Los Menucos (Domnanovich et al., 2008), but the exact stratigraphic repositioning of the material is currently prevented and inherent data are lacking in the literature.

Microfacies characterization

Two thin sections were obtained from the slab MPCA 27029-19 (its footprints are poorly preserved and not included in this study), both parallel and perpendicular to the trampled surface. For the description of the thin sections, Mackenzie, Donaldson & Guilford (1982), and Scasso & Limarino (1997) were taken as a reference. Thin sections are presently stored at the MPCA and labelled as MPCA 27029/19.1 (parallel to the trampled surface) and MPCA 27029/19.2 (perpendicular to the trampled surface).

Measurements

Measurements related to trackmaker body dimension were obtained from slabs MPCA 27029-1, MPCA 27029-9, MPCA 27029-16 and MPCA 27029-21. From single tracks, which are mainly represented by digit traces, measurements of footprint width were taken. Also, track features and differential depth of impressions in some cases allowed to recognize the footprint identity, the side of the trackway when incompletely preserved, or tracks belonging to different trackways (e.g., MMLM 075-1), and element orientations. Track measurements were performed according to guidelines introduced by Leonardi (1987). Track outlines were represented through interpretive drawings.

Digital models

High-resolution digital photogrammetry was undertaken to achieve a more objective representation of track three-dimensional morphology, according to a recently described standard protocol for ichnological studies (Falkingham et al., 2018). To model the studied specimens, the software package Agisoft PhotoScan Pro (Educational License), which enables creating 3D textured meshes by means of semi-automatic processing of images (Mallison & Wings, 2014), was used.

The images selected for the photogrammetric process were acquired using a Nikon Coolpix P520 camera with 4.3–7.6 focal length, resolution 4,896 × 3,672 and pixel size ranging from 1.25 × 1.25 µm and 1.27 × 1.27 µm. Main processing parameters are reported in Table 1. In order to correctly scale the calculated model, a metric reference marker was applied on the surface. Three-dimensional models were converted to colour topographic profiles using the software Paraview (version 5.4.1).

Table 1 Photogrammetric report.

Main processing parameters of the photogrammetric models (from Agisoft Photoscan Professional reports).

3D model	Number of images	Camera altitude (cm)	Ground resolution (mm/pix)	RMS reprojection error	Mean key point size (pix)	Scale bars total error (m)	
MPCA 27029-1	61	55	0.108	0.145637 (0.595447 pix)	3.99373	0.000211567	
MPCA 27029-2	38	49.6	0.142	0.211741 (0.749639 pix)	3.83623	0.000120665	
MPCA 27029-3	36	63.8	0.183	0.221627 (0.663238 pix)	3.1918	0.000101555	
MPCA 27029-4	36	26.2	0.0753	0.225287 (0.695116 pix)	3.21872	0.000145928	
MPCA 27029-5	25	30.2	0.0867	0.186287 (0.548475 pix)	3.41294	0.000125956	
MPCA 27029-9	36	31.9	0.0916	0.206052 (0.638116 pix)	3.27202	0.000154204	
MPCA 27029-16	30	52.3	0.13	0.254353 (0.874324 pix)	3.64063	0.000179922	
MPCA 27029-21	59	47.3	0.127	0.273984 (0.529254 pix)	2.28824	0.000184177	
MPCA 27029-33	74	50.2	0.144	0.25616 (0.993387 pix)	3.90857	0.000121254	
MMLM 075-1	54	37.8	0.0898	0.196238 (0.739726 pix)	3.75732	0.000118491	
MMLM 1	77	42.5	0.101	0.222075 (0.673395 pix)	3.09592	0.00332392	
MMLM 2	52	33	0.0949	0.234591 (0.852769 pix)	3.78559	5.99994e–05	

Geological Setting

Continental deposits of Triassic age in Argentina were accumulated in different basins in western and northwestern regions (Mendoza, San Juan, San Luis and La Rioja provinces) as well as in Patagonia (northern sector of the Santa Cruz province and Río Negro provinces). These elongated, narrow rift basins with prevalent NW-SE and NNW-SSE trends were developed during Permian and Triassic periods and are related with the breakup of the western margin of south-west Gondwana (Kokogian et al., 1999; Franzese & Spalletti, 2001; Barredo et al., 2012).

The Triassic tetrapod track record of southern South America is exclusive to three basins, namely the Ischigualasto-Villa Unión Basin (San Juan and La Rioja provinces), the Cuyo Basin (Mendoza and San Juan provinces) and the Los Menucos basin (Río Negro province) (e.g., Melchor, Genise & Poiré, 2001; Melchor & de Valais, 2006; de Valais, 2008 and references therein). According to Spalletti (1999), in the northern basins (i.e., Ischigualasto-Villa Unión and Cuyo) the sedimentation encompasses the Lower to Upper Triassic, while in the Los Menucos Basin the sedimentation took place in the Late Triassic, based on the age of volcanic activity in north-central Patagonia.

After the works of Stipanicic (1967), Stipanicic et al. (1968) and Stipanicic & Methol (1972), Stipanicic & Methol (1980), the Los Menucos Group (also as ‘Complejo Los Menucos’—Los Menucos Complex sensu Cucchi, Busteros & Lema, 2001) was established by Labudía & Bjerg (2001) to indicate dacitic to rhyolitic ignimbrites, mesosilicic lavas and subordinate Triassic sedimentary rocks exposed around Los Menucos town, in the north-western sector of the North Patagonian Massif (Río Negro province, Argentina; Fig. 1A).

Figure 1 The Los Menucos area.

(A) Location map and geological sketch of Los Menucos area (from Labudía & Bjerg (2005), redrawn and slightly modified). White star indicates Estancia Yancaqueo, from which the Pentasauropus footprints come. (B) Simplified stratigraphic section of the Los Menucos Group (from Labudía & Bjerg (2005), redrawn and slightly modified). Dashed lines with quotation marks indicate the possible position of Pentasauropus-bearing strata.

Within the Los Menucos Group, two lithostratigraphic units were defined, namely the Vera Formation at the base and the Sierra Colorada Formation on top (Labudía & Bjerg, 2001; Labudía & Bjerg, 2005, and references therein; Fig. 1B). The Vera Formation, from which tetrapod tracks are historically reported, is mainly composed of volcanic and continental deposits laid down inside small basins bordered by regional and local faults with strike NE-SW, E-W and NW-SE (Labudía & Bjerg, 2001; Labudía & Bjerg, 2005). The Vera Formation is mainly represented by brownish to yellowish conglomerates, white to greenish sandstones and reddish brown to red pelites, with which volcanic ashes, tuffs and tuffites, dacitic pyroclastic flow products and volcanic breccias are intercalated (Labudía & Bjerg, 2001; Labudía & Bjerg, 2005). Sedimentation took place mainly in alluvial plain, floodplain, ephemeral river and small lacustrine palaeoenvironments (Labudía & Bjerg, 2005), under seasonal climate condition with alternating periods of dry and wet conditions (Gallego, 2010). Sedimentary and volcaniclastic levels within the Vera Formation are characterized by a very rich palaeoflora, the so-called “Dicroidium - type flora” (Stipanicic, 1967; Stipanicic & Methol, 1972; Artabe, 1985a; Artabe, 1985b; Labudía et al., 1995; Labudía & Bjerg, 2001; Labudía & Bjerg, 2005) and by an abundant tetrapod ichnofauna, preserved on sandstones with poorly sorted grains and with a variable content of tuffaceous breccias (Melchor & de Valais, 2006). Finds of skeletal fauna are scarce and are so only represented by remains of an amiiform fish (Bogan, Taverne & Agnolin, 2013).

The Sierra Colorada Formation is essentially made of ignimbritic volcanic rocks (Labudía & Bjerg, 2001; Labudía & Bjerg, 2005), dated at 222 ± 2 Ma with the Rb/Sr isochron method (Norian, Late Triassic; Rapela et al., 1996) and at 206.9 ± 1.2 Ma with the Ar/Ar method (Rhaetian, Late Triassic; Lema et al., 2008). These datations do not radiometrically constrain the base of the Vera Formation, for which a Late Triassic age was historically proposed on the basis of the “Dicroidium-type flora” and the tetrapod ichnofauna.

More recent results indicated an age of 257 ± 2 Ma (Wuchiapingian, Late Permian) for a rhyolitic ignimbrite, 252 ± 2 Ma (Changhsingian, Late Permian) for an andesite, and 248 ± 2 Ma (Olenekian, Early Triassic) for a dacitic ignimbrite (Luppo et al., 2017) of the Los Menucos Group. These new data predate the main volcanic activity to an about a 10 Ma period between the Late Permian and the Early Triassic, making the lower part of the Los Menucos Group coeval with the La Esperanza Plutono-Volcanic Complex (González et al., 2017; Luppo et al., 2017).

Sedimentological observations

Track-bearing slabs consist of yellowish to greenish, medium to mainly coarse grained and poorly sorted volcaniclastic sandstone lacking of sedimentary structures in hand samples, neither on the surface or cross-section.

The observed texture ranges from inequigranular/equigranular (Figs. 2A, 2B) to predominantly equigranular (Fig. 2C). Phenocrysts, mainly subhedral and anhedral, range in dimension from 0.5 mm to 1.5 mm and show in one case incipient orientation. Phenocrysts are represented mainly by plagioclase, quartz, alkaline feldspar, biotite, amphibole (hornblende), orthopyroxene (enstatite) and calcite floating in a mafic, glassy matrix.

Figure 2 Thin sections (MPCA 27029/19.1 and MPCA 27029/19.2) of track-bearing slab MPCA 27029-19.

Inequigranular, epiclastic texture with anhedral and subhedral phenocrysts at the base (A) and middle portion (B) of the track-bearing slab MPCA 27029-19. (C, D) Equigranular less epiclastic texture indicating a minor sedimentary reworking of the trampled surface.

The dominant epiclastic texture observed at the base of the trampled surface (thin section MPCA 27029/19.2), mainly represented by fragments of quartz and some lithics displaying attrition and rounded to sub-angular shape, suggesting sedimentary reworking of an original tuff of probable dacitic composition. The texture observed in the thin section MPCA 27029/19.1 instead indicates a limited sedimentary reworking (Fig. 2D). In section, a faint normal gradation can be observed most likely indicating short sedimentation events; on the whole, the track-bearing slabs can be related to a proximal fluvial environment.

Track Record

Track preservation

Specimen MMLM 075-1 is composed of four slabs, two as negative (concave) relief, labeled as MMLM 075-1/1a, /2 and /3a, and two as their positive (convex) filling, labeled as MMLM 075-1/1b and 3/b. There are no evidences of any layer between the concave and the convex reliefs and the shape of both concave epireliefs and convex hyporeliefs are exactly complementary (Fig. 3). Therefore, and taking into account that the tracks preserved similarly (i.e., sub-circular/sub-ovoidal to pointed digit impressions; roughly sub-circular to elliptical pad tracks; very thin displacement rims in the pad and well-marked in the digit impressions; Figs. 4–9), in our opinion the concave epireliefs are true tracks (sensu Marty, Falkingham & Richter, 2016) and the convex hyporeliefs are their natural casts (sensu Marty, Falkingham & Richter, 2016).

Figure 3 Tracks mode of preservation.

Convex hyporeliefs (A, C) fitting with concave epireliefs (B, D) preserved on slab MMLM 075-1 (true tracks and natural casts, respectively).

Figure 4 Photos, three-dimensional models and interpretative drawings of the studied material.

(A) Track-bearing slabs MPCA 27029-1; (B) solid three-dimensional model of (A); (C) colour topographic profile and (D) interpretative drawing of (A). (E) Track-bearing slabs MPCA 27029-2; (F) solid three-dimensional model of (E); (G) colour topographic profile and (H) interpretative drawing of (E). In (A)–(D), footprint 2 and 5, note the non-impressed area between the sole pad trace and the base of digit traces. In (E)–(H) note the displacement areas behind digit traces, interpreted as the result of the thrust of digit pushing the sediment backwardly.

In general, the tracks studied here are moderately well preserved (grade 1 sensu Belvedere & Farlow, 2016), and the true tracks are not elite tracks (sensu Lockley, 1991). In addition, they are not modified true tracks (sensu Marty, Falkingham & Richter, 2016) because they lack evidence of physiochemical (e.g., weathering) and/or biological influences after they were made. Thereby, the shape of these tracks is mainly conditioned by the substrate consistency (grain size and water content). Recently, Falk et al. (2017) performed neoichnological experiments that compared the shape of tracks impressed in three different sediments (fine, medium and coarse sand) with different moisture contents (wet, moist and dry). They concluded that wet and dry coarse sediments preserve tracks without fine details, but moisture coarse sediment might preserve the overall track shape and details as claw impessions. As has been previously commented, the tracking surface is a medium to coarse sandstone, and tracks have depth digit impressions with extruded rims.

Therefore, and according to the Falk et al. (2017)’s experiments, the trackmakers most likely walked on humid, not waterlogged nor dry, coarse sediments with a moderately plastic behaviour, able to record the main anatomical features of the autopods.

Track description

The material are manus and pes tracks with very low dimensional heteropody (i.e., condition in which the autopods are dimensionally and morphologically different), mainly preserved as tetradactyl impressions, although pentadactyl tracks are also present (MPCA 27029-1/4/6, MPCA 27029-2/2, MPCA 27029-4/2, MPCA 27029-16/10, MPCA 27029-33/2, MMLM 075-1) (Figs. 4, 5E–5H, 7A–7D, 8), as well as tridactyl ones displaying only the central digits (MPCA 27029-9/2, MPCA 27029-16/8, MPCA 27029-21/3, MMLM 1/1, MMLM 2/3) (Figs. 6G–6H, 7, 9). Morphologically, manus and pes tracks are strongly symmetrical. Digit traces are commonly arranged to shape an arcuate pattern that is convex anteriorly, according to which the digit III trace (the central one) or digit III and IV traces are the most projecting. Variability affecting the number of digits can occur on the same slab (e.g., MPCA 27029-21, MMLM 2; Figs. 7E–7H, 9E–9H ). In the material under study the degree of curvature of the arcuate pattern is variable and appears more pronounced in some smaller tracks (e.g., MPCA 27029-16/7/9/10; Figs. 7A–7D) than in larger ones (e.g., MPCA 27029-1/4, MMLM 075-1, MMLM 2/2; Figs. 4, 8E–8H, 9E–9H). In the smaller tracks (e.g., MPCA27029-16, interpreted as left by a juvenile individual), the morphology of digit traces, their relative spacing and orientation, as well as the position of pes and manus impression is comparable with that of the larger tracks. When present, also the sole pad trace resembles that observed in the footprints of larger dimension. Thus, apart from the degree of curvature, the general morphology remains consistent despite dimensional differences (see Figs. 4A–4D and 7A–7D).

Figure 5 Photos, three-dimensional models and interpretative drawings of the studied material.

(A) Track-bearing slabs MPCA 27029-3; (B) solid three-dimensional model of (A); (C) colour topographic profile and (D) interpretative drawing of (A). (E) Track-bearing slabs MPCA 27029-4; (F) solid three-dimensional model of (E); (G) colour topographic profile and (H) interpretative drawing of (E). Note the digit trailing marks slightly affecting the digit traces of footprint 2 in (A)–(D), which are absent in footprints showed in (E)–(H) where digit traces are roughly sub-circular in morphology.

Figure 6 Photos, three-dimensional models and interpretative drawings of the studied material.

(A) Track-bearing slabs MPCA 27029-5; (B) solid three-dimensional model of (A); (C) colour topographic profile and (D) interpretative drawing of (A). (E) Track-bearing slabs MPCA 27029-9; (F) solid three-dimensional model of (E); (G) colour topographic profile and (H) interpretative drawing of (E).

Figure 7 Photos, three-dimensional models and interpretative drawings of the studied material.

(A) Track-bearing slabs MPCA 27029-16 produced by a juvenile trackmaker; (B) solid three-dimensional model of (A); (C) colour topographic profile and (D) interpretative drawing of (A). (E) Track-bearing slabs MPCA 27029-21; (F) solid three-dimensional model of (E); (G) colour topographic profile and (H) interpretative drawing of (E). The general morphology and structure of footprints 6–10 in (A)–(D), left by a juvenile trackmaker, is identical to that characterizing larger footprints even when preserved only as digit traces.

Figure 8 Photos, three-dimensional models and interpretative drawings of the studied material.

(A) Track-bearing slabs MPCA 27029-33; (B) solid three-dimensional model of (A); (C) colour topographic profile and (D) interpretative drawing of (A). (E) Track-bearing slabs MMLM 075-1; (F) solid three-dimensional model of (E); (G) colour topographic profile and (H) interpretative drawing of (E). In (E)–(H) note the long and sharp digit trailing marks.

Figure 9 Photos, three-dimensional models and interpretative drawings of the studied material.

(A) Track-bearing slabs MMLM 1; (B) solid three-dimensional model of (A); (C) colour topographic profile and (D) interpretative drawing of (A). (E) Track-bearing slabs MMLM 2; (F) solid three-dimensional model of (E); (G) colour topographic profile and (H) interpretative drawing of (E). In (A)–(D) note the long and sharp digit trailing marks affecting footprints 2 and 4, resembling those of Figs. 8E–8H.

Digit traces can be characterized by a sub-circular/sub-ovoidal morphology (e.g., MPCA 27029-9, MPCA 27029-16, MPCA 27029-21; Figs. 6E–6H, 7), while in other cases they are markedly pointed (e.g., MPCA 27029-2, MPCA 27029-4, MPCA 27029-5, MPCA 27029-33; Figs. 4E–4H, 5E–5H, 6A–6D, 8A–8D). These two morphologies can co-exist on the same slab and within the same set or trackway, thus pertaining to the spectrum of internal variability of the material under study. When pointed, the most medial digit traces (i.e., digit I or II imprints and, to a lesser extent, digit III and IV imprints), both of manus and pes tracks, can be affected by drag marks. These extramorphological features (see Peabody, 1948) qualitatively range from weakly hinted and short (e.g., MPCA 27029-1/6, MPCA 27029-2/1, MPCA 27029-3/1/2, MPCA 27029-4/2, MPCA 27029-5, MPCA 27029-9/4/6; Figs. 4–6) to highly sharp and long (e.g., MMLM 075-1, MMLM 1 and MMLM 2; Figs. 8E–8H, 9A–9D).

Central digits are commonly the most deeply and uniformly impressed, both in manus and pes tracks (e.g., MPCA 27029-4, MPCA 27029-5, MMLM 1; Figs. 5F–5G, 6B–6C, 9B–9C). When a certain degree of variability is observed, digit III and IV imprints are the most deeply imprinted (e.g., MPCA 27029-1, MPCA 27029-3, MPCA 27029-16; Figs. 4B–4C, 5B–5C, 7B–7C), followed by digit II and I imprints. The digit V trace, when preserved, is shorter and closer to the pad trace than the other digit traces and is only faintly imprinted (e.g., MMLM 075-1, Figs. 8E–8H; but see MPCA 27029 − 16∕9∕10 for a different configuration of the digit depth of impression, most likely due to the reaction of the substrate and water content of the sediments at the time of impression, Figs. 7B–7C).

Behind the digit traces, a roughly sub-circular to elliptical sole pad trace can be preserved (e.g., MPCA 27029-1/2/4/5/6, MPCA 27029-5, MPCA 27029-16/10, MMLM 075-1, MMLM 1/2/4, MMLM 2/1; Figs. 4A–4D, 6A–6D, 7A–7D, 8E–8H, 9). The sole pad trace lies at a short distance from the base of the central digit traces and commonly approximates the most medial and lateral digit imprints (e.g., MPCA 27029-1; Figs. 4A–4D). Commonly, the sole pad trace is separated from central digit traces ahead by a non-impressed area, which appears as a groove or as a ridge depending on the mode of preservation, tapering towards the most medial and lateral digit imprints. This should not be confused with displacement areas of similar morphology, which are instead related to digit traces (i.e., thrust of digit pushing the sediment backwardly; Fig. 10), where this area is not impressed (e.g., MPCA 27029-2, MPCA 27029-3/1/4, MPCA 27029-4, MPCA 27029-9/3/5, MPCA 27029-21/4, MPCA 27029-33/1/2/3, MMLM 2/3; Figs. 4E–4H, 5, 6E–6H, 7E–7H, 8A–8D, 9E–9H).

Figure 10 Morphological and extramorphological features identified on the studied material.

(A) Manus track MPCA 27029/2 and (B) interpretative drawing. (C) Pes track MPCA 27029-1/5 and (D) interpretative drawing. Extramorphological features are in blue and grey; morphological features are in black.

The sole pad trace is more deeply impressed in its central portion; depth of impression slightly decreases toward the lateral and distal portion (i.e., close to the non-impressed area behind digit traces, MPCA 27029-1, MMLM 075-1, MMLM 1;  Figs. 4B–4C, 8F–8G, 9B–9C).

When possible, we tried to define the orientation of the footprint axis with respect to the trackway midline. The axis of pes tracks is in some cases rotated inwardly with respect to the trackway midline but it can also be parallel to the trackway midline (e.g., MPCA 27029-9, 27029-16), while manus tracks show a wider range of variability, being both inwardly and outwardly rotated with respect to the hypothetical trackway midline (e.g., MPCA 27029-1 and MMLM2, respectively). When possible, measurements and ratios were taken; measurements were performed taking into account digit III as the homologous point, both for manus and pes tracks. Results are reported in Tables 2 and 3. In general, footprints are wider than long with oblique pace length ranging between 60% and 80% of the stride length. Pace angulation ranges from 70° to 101°. Gleno acetabular distances, measured considering an amble gait (other gaits are reported in Table 2), indicate trackmakers with trunk length approximately of 37.4 cm (MPCA 27029-16), 63.1 cm (MPCA 27029-9) and 91.6 cm (MPCA 27029-1). The proximal margin of the digit traces lies at less than 3 cm from the distal margin of the sole trace (Table 3).

Table 2 Mean measurements (in cm) of track and trackway parameters.

Specimen	Fl	Fw	Mpl	Ppl	Mpa	Ppa	Msl	Psl	ETW	ITW	ETW/SL	GAD	Psl/GAD	
MPCA 27029-1	10.6
12.8
8.9	10.3
12.2	42.2
33.3	41.7
40.5	101°	99°	59.5	62.4	50.0	3.65	0.82	46.2 (a)
47.8 (b)
91.6 (c)	1.01	
MPCA 27029-2	/	15.7
13.5	/	/	/	/	/	/	/	/	/	/	/	
MPCA 27029-3	/	12.7
14.7	/	/	/	/	/	63.5	/	/	/	/	/	
MPCA 27029-4	/	12.6
13.3	/	/	/	/	/	/	/	/	/	/	/	
MPCA 27029-5	6.6	11.6	/	/	/	/	/	/	/	/	/	/	/	
MPCA 27029-9	/	7.4
10.9	28.5	37.0
32.5	/	81°	/	45.5	36	7.36	0.79	40.0 (a)
46.9 (b)
63.1 (c)	0.91	
MPCA 27029-16	5.6	8.1
7.1	21.0
15.0	22.5	100°	/	28.5	/	23.0	4.40	0.81	26.0 (a)
33.1 (b)
37.4 (c)	/	
MPCA 27029-21	/	10.7	/	37.0
34.0
34.5	/	70°	/	37.5
41	38.5	5.85	1.03	/	/	
MPCA 27029-33	/	15.4
12.8	/	/	/	/	/	63.5	/	/	/	/	/	
MMLM 075-1	10.2
11.8	13.6
9.8	/	/	/	/	/	57.3	/	/	/	/	/	
MMLM 1	10.2
10.9	13.5
14	/	/	/	/	52.0	/	/	/	/	/	/	
MMLM 2	/	12.2
11.4	/	/	/	/	40.5	/	/	/	/	/	/	
Notes.

ETW external trackway width

Fl footprint length

Fw footprint width

GAD gleno-acetabular distance: (a), ‘primitive’ alternate pace (the trunk length of the producer is underestimated); (b) alternate pace; (c) amble (a, b, c, considering primary overlap sensu Leonardi, 1987)

ITW internal trackway width

Mpa manus pace angulation

Mpl manus pace length

Msl manus stride length

Ppa pes pace angulation

Ppl pes pace length

Psl pes stride length

Psl/GAD pes stride length/gleno-acetabular distance ratio

ETW/SL external trackway width/stride length ratio

Table 3 Sole pad-ungual trace distance.

Distance (in cm) between the distal margin of the sole pad trace and the proximal margin of the digit traces in complete Pentasauropus footprints. The measurements most likely indicate a raised and inclined position of the metapodial elements of fore and hind foot in the Pentasauropus trackmaker.

	Digit I	Digit II	Digit III	Digit IV	Digit V	
MPCA 27029-1	
Footprint 2	1.35	1.41	1.84	1.22	/	
Footprint 4	1.68	1.66	1.84	1.91	/	
Footprint 5	/	1.26	1.45	1.52	1.20	
Footprint 6	1.2	1.25	2.18	2.24	1.33	
MPCA 27029-5	
Footprint 1	/	0	0.96	1.13	0.4	
MPCA 27029-16	
Footprint 10	0.92	0.9	1.1	0.94	0.79	
MMLM 075-1	
Footprint 1	1.65	2.09	2.94	2.56	1.45	
Footprint 3	/	1.41	1.68	2.19	1.63	
Footprint 4	1.95	2.25	2.45	2.25	1.58	
MMLM 1	
Footprint 2	1.88	2.53	2.73	2.25	/	
Footprint 4	1.59	1.86	2.25	2.12	/	
MMLM 2	
Footprint 1	1.66	2.37	2.02	2.27	/	

Remarks

The footprints from the Los Menucos ichnosite are characterized by having the following features: homopodic manus and pes tracks with low dimensional heteropody, up to five digit imprints aligned, forming an anteriorly convex arch, a sole pad trace more impressed centrally or centro-laterally. On the basis of these general features, the specimens from Los Menucos are tentatively referred to as Pentasauropus.

The ichnogenus Pentasauropus (Ellenberger, 1970) was established on the basis of material collected and described years before (Ellenberger, 1955) from the Upper Triassic lower Elliot Formation (Stormberg Group) of the Karoo Basin of Lesotho (Southern Africa). Five ichnospecies were originally included in the ichnogenus, namely Pentasauropus erectus, Pentasauropus incredibilis, Pentasauropus maphutsengi, Pentasauropus morobongensis and Pentasauropus motlejoi, which remained unchanged in the subsequent formal listing (Ellenberger, 1970; Ellenberger, 1972). Material from the Ellenberger collection referred to this ichnogenus is housed at the University of Montpellier (France) and represented by six casts originally mentioned as Pentasauropus incredibilis (LES 054 1-3, LES 054 4), Pentasauropus morobongensis (LES 005) Tetrasauropus gigas (LES 038), plus some missing specimens (see D’Orazi Porchetti & Nicosia, 2007, and reference therein for a complete assessment of inventory numbers).

After the original and subsequent publications of Ellenberger (1955), Ellenberger (1970), Ellenberger (1972), the ichnogenus was considered as valid by Olsen & Galton (1984), Lockley & Meyer (2000), D’Orazi Porchetti & Nicosia (2007), Bordy, Abrahams & Sciscio (2017), and Hunt, Lucas & Klein (2018). D’Orazi Porchetti & Nicosia (2007) emended the diagnosis of the ichnogenus to appoint the type ichnospecies and considered the five ichnospecies as synonyms of Pentasauropus incredibilis. Differences in track pattern were considered as originated by dimensional constraints and/or behavioural factors, and the main footprint characteristics (e.g., number and arrangement of digits, heteropody) do not justify an ichnospecies separation (D’Orazi Porchetti & Nicosia, 2007). Moreover, agreeing with Lockley & Meyer (2000), the same authors assigned tracks originally referred to as Tetrasauropus gigas to Pentasauropus.

In agreement with the emended ichnogeneric diagnosis by D’Orazi Porchetti & Nicosia (2007), the arcuate pattern of manus and pes tracks derived from the five equally spaced claw or ungual traces (those of imprints of digit II, III and IV are the largest). In other cases a roughly rounded sole pad is observed behind claw or ungual traces (LES 053 A, B, C in Ellenberger, 1972, pl. IV and V, and LES 038). According to D’Orazi Porchetti & Nicosia (2007), the axis of the pes impressions is always inwardly rotated, while that of the manus impression can range from slightly inwardly rotated (LES 052 B and LES 053 A) to slightly outwardly rotated (LES 038 and LES 052 A). Although long and complete trackways are not represented, this feature seems to characterize also the studied material based on the reconstruction of an hypothetical midline (e.g., Figs. 4A–4D, footprints 4 and 6; Figs. 5A–5D, footprint 2). In some cases (e.g., MPCA 27029-9 and MPCA 27029-21, Table 2) short stride length in relation to overall footprint dimension could indicate a primary overstepping. However, in our opinion this trackway characteristic cannot be ensured on the basis of the material under study and most probably, taking into account the complexity of the dynamic of locomotion, cannot be inferred only from the stride lengths and footprint dimensions. Thus, for the time being, we prefer not to stress the interpretation about the overstepping. Additionally, overprinting in the studied material is absent; in some cases (e.g., MPCA 27029-3 and MPCA 27029-33, Figs. 5B, 8B) interferences indicate that the hind print in the set was left after the fore print. For these cases, overstepping is not sustainable.

Outside of the Southern Africa, tracks tentatively referred to the ichnogenus were reported from Upper Triassic Chinle Group of Utah (Lockley & Hunt, 1995; Hunt-Foster et al., 2016) and Colorado (Gaston et al., 2003, Fig. 12B), both USA. Moreover, tracks possibly referable to Pentasauropus were found in the Gettysburg Shale of the Gettysburg Basin of the Newark Supergroup (Baird pers. comm in Olsen & Galton, 1984). In Argentina, apart from the report from the Triassic Vera Formation (Río Negro province, Domnanovich et al., 2008), tracks referred to as Pentasauropus were described from the Carnian Portezuelo Formation (‘Type Q2’ sensu Marsicano & Barredo, 2004). In addition, tracks with similar morphology to Pentasauropus were also reported from the Middle Triassic Cerro de Las Cabras Formation (Mendoza province, as cf. Pentasauropus in de Valais, Melchor & Bellosi, 2006) and from the Portezuelo Formation (San Juan province) as ‘huellas cuadrúpedas tipo C’ (i.e., quadrupedal tracks type C; de Valais, 2008) but, for the time being, this material remains in open nomenclature.

Zoological attribution

Several attempts to identify the trackmaker of Pentasauropus have been made. The ichnogenus was originally attributed to amphibians, basal melanorosaurid, ornithischian, anapsid and basal sauropod (Ellenberger & Ellenberger, 1958: p. 67; Ellenberger, 1970; Ellenberger, 1972). Moreover, Haubold (1974); Haubold (1984) referred Pentasauropus to a sauropod or therapsid trackmaker. A dicynodont was also proposed as producer by Olsen & Galton (1984), Anderson, Anderson & Cruickshank (1998) and Lockley & Meyer (2000). Galton & Heerden Van (1998) attributed Pentasauropus to large anomodont dicynodonts. D’Orazi Porchetti & Nicosia (2007) accepted the attribution to a dicynodont, observing a good match between the skeletal autopodia of Triassic dicynodonts and the structure of digital impressions of the manus and pes, apart from the strong homopody and the limb posture (see also Walter, 1986). Recently, the kannemeyeriiform dicynodont Pentasaurus goggai from the lower Elliot Formation of South Africa has been referred as probable trackmaker of Pentasauropus tracks from the same lithostratigraphic unit (Kammerer, 2018).

The studied material from the Los Menucos locality presents some features that allow corroboration of the therapsid interpretations about trackmaker identity. At the same time, constraining the identity of the putative trackmaker opens the way for new inferences about the posture of the autopodia.

Limb posture kept by Pentasauropus trackmakers during the step cycle can be tentatively inferred from the trackway pattern (Peabody, 1948; Peabody, 1959; Kubo & Benton, 2009; Kubo & Ozaki, 2009), even if this interpretation is often far from being simple and linear (Crompton & Jenkins Jr, 1973). For example, it must be noted that non therapsid-synapsids with sprawling posture could have left trackways in which the left and right tracks lie near to the axial midline (i.e., narrow internal trackway width mirroring a semi-erect to erect posture of the trackmaker) by adopting side to side flexion of the trunk (Hopson, 2015). The same was described for a therapsid trackway by Smith (1993). Some degree of lateral undulation of the vertebral column causing swing of the hips has been also described on the basis of skeletal remains (Fröbisch, 2006: p. 1305). However, in sprawling trackmakers adopting trunk flexion and producing narrow trackways, tracks are mainly inwardly oriented with respect to the trackway midline (see Hopson, 2015: Fig 8.1). In the studied material, the orientation of footprint axis (passing through digit III) is parallel to the travel direction. This feature, combined with the extremely narrow internal trackway width measured from pes tracks (see Table 2 and Figs. 4A–4D, 5A–5D, 6E–6H, 7A–7D), allow to exclude a sprawling posture and most likely indicate a semi-erect posture for the trackmaker hind limbs. A lateral trunk undulation during the step cycle could have been also adopted by the Pentasauropus trackmaker and it would account for the variable pace angulation measured in manus tracks, possibly coupled with low trackmaker velocity.

A more upright posture with respect to that of non-therapsid synapsids is also indirectly sustained by the symmetry of manus and pes tracks. This feature mirrors a symmetry of the trackmaker’s autopods, a feature combined with the acquisition of upright posture and limbs parallel to the sagittal plane of the trackmaker during locomotion (Romer, 1956; Hopson, 1995).

Number of digit imprints, symmetry of manus and pes track, and the morphology of unguals enable us to corroborate previous interpretations and suggest a dicynodont as the most probable trackmaker of Pentasauropus. Also the limb posture as supposed from tracks sufficiently matches those discussed for Triassic dicynodonts by Fröbisch (2006).

Pentasauropus producers, based on the type specimens, were characterized by a very low dimensional heteropody and morphological homopody. Thus, mainly based on these characters, a quite confident match can be found, among dicynodonts, with Kannemeyeriiformes (see, for example, the descriptions and reconstructions of the autopods of Dinodontosaurus by Morato (2006: Fig. 30), Tetragonias njalilus by Cruickshank (1967: Fig. 17) and by Fröbisch (2006: Fig. 9). Kannemeyeriiform manual and pedal skeletal elements are in fact generally homomorphic and conservative into the clade, as recently stated by Kammerer (2018). Variation in manus morphology among kannemeyeriiforms is only limited to minor differences in ungual shape (Lucas, 2002). Digit traces are here considered roughly compatible with broad ungual phalanges characterized by rounded tips but the exact shape cannot be determined. The morphological variability of ungual traces most likely relates to substrate conditions at the time of impression and, as discussed below, from the dynamics of the locomotion of the producers.

Among the other possible producers already mentioned in the literature, are excluded: (i) an amphibian trackmaker, because of the pentadactyl manus and general morphology; (ii) an anapsid trackmaker for the trackway configuration and footprint axis orientation; (iii) a sauropodomorph and sauropod trackmaker for the trackway configuration, footprint axis orientation and morphological homopody.

The posture of the autopodia of the Pentasauropus trackmaker that can be inferred from the ichnological material differs from that inferable from the description by Cruickshank (1967). In those Pentasauropus tracks that show an impression of the sole or palm, a negligible distance between the distal margin of the sole/palm pad trace and the proximal margin of the central digit traces, further reducing towards digits I and V, was observed (Table 3). This feature most likely indicates that not all of the foot bones contacted the ground during locomotion, at the same time constraining the orientation of metapodial and basipodial elements, and also that of the more proximal phalanges, in the articulated autopod. Thus, the reconstruction proposed here contemplates an inclined position of pedal and manual elements in the autopods of Pentasauropus producers. The sub-circular to elliptical pad trace behind the digit traces is consequently considered compatible with an extended fleshy pad below the basipodials and likely metapodials of the autopods of the producer (Fig. 11).

Figure 11 Limb and autopod posture in the Pentasauropus trackmaker.

Simplified reconstruction of limb posture in back (A) and lateral (B) views. Simplified reconstruction of zeugopodials and hind autopod in lateral (C) and bottom (D) views. In colour the possible extension of the fleshy cushion on which the basipodials rested, ensuring support during locomotion. See the supplementary video to get a more complete view of the reconstruction. Artwork by Fabio Manucci.

Discussion

The autopod posture of Pentasauropus trackmaker

Compared to the ichnogenus Pentasauropus, tracks from Los Menucos Group have allowed us to verify previous ichnological interpretations based on the reference material from Lesotho and have enabled us to corroborate the identification of a putative trackmaker and its limb posture. Moreover, the studied material sheds light on the trackmaker autopod posture. The smaller tracks (e.g., MPCA27029-16) are here interpreted to have been left by a juvenile trackmaker and allowed to appreciate that track morphology and structure are uniform in different ontogenetic stages of the same type of producer.

Number of digit imprints, symmetry of manus and pes track, morphology of ungual traces, limb posture and morphologically homopodic manus and pes tracks indicate the producer of Pentasauropus to be sought among dicynodonts of the clade Kannemeyeriiformes (Fröbisch, 2009). Within this panorama and accepting the proposed palaeozoological attribution, Pentasauropus tracks represent a valuable datum, further enriching the dicynodont record of the Triassic of Argentina (Cox, 1962; Bonaparte, 1969; Bonaparte, 1971; Bonaparte, 1981; Lucas, 1998; Lucas, 2010; Lucas, 2018; Rogers et al., 2001; Zavattieri & Arcucci, 2007; Fröbisch, 2009; Domnanovich & Marsicano, 2012; Abdala et al., 2013; Mancuso et al., 2014).

The studied tracks enabled us to improve the knowledge of the therapsid faunas from south-western Gondwana, especially about their locomotion and functionality of fore and hind autopods. The inferred limb posture of the Pentasauropus trackmaker finds a match with the osteological data provided by the therapsid record (e.g., King, 1981a; Fröbisch, 2006) and allows to corroborate interpretations derived from body-fossils. Meristic and qualitative track characteristics and trackway parameters, if jointly considered, suggest that the Pentasauropus trackmaker had a semi-erect to erect posture, especially the hind limbs (Figs. 11A–11B).

Contrarily to what was stated about therapsid posture in the past (Charig, 1980; Bonaparte, 1982), therapsid-grade limb osteology was characterized by several important modifications, which indicate a more parasagittal stance of the limbs (Romer, 1956; Boonstra, 1967; Jenkins Jr, 1971), especially if compared with the prevalent sprawling posture of non-therapsid synapsids (Romer, 1956; Hopson, 2015). Modifications of the scapula and the glenoid have allowed the elbow to rotate inwardly, bringing the humerus closer to the sagittal plane (Walter, 1986). The iliac blade was expanded anteriorly and allowed the insertion of a larger iliofemoralis muscle (i.e., the muscle that allows movement of the femur with respect to the hip), enabling femoral retraction (Romer, 1956; Walter, 1986). Moreover, the femoral head folded medially and enabled a more parasagittal position of the propodial (Romer, 1922; Walter, 1986).

Concerning the dynamics of locomotion in non-mammalian therapsids, Kemp (1978) proposed a dual-gait condition, intermediate between the plesiomorphic gait of amniotes (Sumida & Modesto, 2001) and the mammalian erect gait, based on the therocephalian Regisaurus jacobi. This condition has proved to be not possible for derived dicynodonts, such as Kingoria nowacki (sensu King, 1985) and for kannemeyeriiformes dicynodonts, all characterized by an ankle joint inhibiting extensive rotational movements needed for dual-gait locomotion (Fröbisch, 2006). In dicynodonts, the forelimb step cycle was performed in an abducted (i.e., sprawling) posture, whereas the hind limb step cycle passed from a primitive abducted posture in earlier members, such as Robertia broomiana (see King, 1981b) to an adducted (i.e., erect) posture in more derived taxa (Walter, 1986), such as Dicynodon trigonocephalus and Tetragonias njalilus (e.g., King, 1981a; Fröbisch, 2006).

The autopod posture proposed for the studied tracks quite differ from the information and reconstructions derived from the body-fossil record. As stated before, the alleged autopodial structure inferred from Pentasauropus tracks is dictated by the relative distance between the proximal portion of digital (ungual) traces and the distal edge of the sub-circular pad trace, which has been inferred to relate to a fleshy pad behind the basipodial. The observed track morphology seems to imply that, except for acropodial and the fleshy pad, no other bony elements of the producer’s autopods were imprinted on the substrate, indicating that they were likely raised in position. Such a configuration is considered valid for the foot bones in a static-state and would fall at least within a subunguligrade posture, implying that the phalanges were the only bony pedal elements contacting the ground in a static stance. However, if the three-dimensional footprint morphology is considered (i.e., ungual traces and pad trace behind them) concurrently with spatial data regarding pad trace/digit trace distance (Table 2), it is evident that the unguals were not the only pedal elements performing the cycle of locomotion. Thus, the foot cannot be regarded as subunguligrade from a dynamic point of view. During locomotion, the body weight of Pentasauropus producers was not carried only by phalanges but, most likely, the entire foot supported the load (Figs. 11 and 12). The fleshy pad behind the basipodials actively contacted the ground most likely during the touch-down and weight-bearing phase, as was already inferred from footprint depth of impression in other producers (e.g., Romano, Citton & Nicosia, 2016; Citton et al., 2017). Thus, from a functional standpoint, the autopod posture of the Pentasauropus trackmaker can be regarded as plantiportal (sensu Michilsens et al., 2009) and it is considered to have been maintained during different gaits. Such a posture could have been accompanied by an arched configuration of the articulated metapodials and at least of the proximal phalanges (Kümmell & Frey, 2012) (Figs. 11C– 11D). Metacarpals forming an arched configuration when articulated were described in a specimen of Tetragonias njalilus (Cruickshank, 1967), and this kind of configuration could have been accompanied by little movement capabilities (Rubidge & Hopson, 1996) of the metapodials and could have dictated the observed relative position of the ungual traces. A manual/pedal structure like the one here hypothesized could have maintained a large surface in contact with the ground by means of cartilaginous elements and fleshy cushions on which the basipodials rested, ensuring a supportive role of the whole autopods during the cycle of locomotion and particularly during the maximum load. Digit traces were formed by phalanges deeply penetrating into the substrate during the final weight-bearing phase, kick-off and thrust. This could explain the different depth of the impression that is observed in completely preserved tracks. Among digits the series II-IV, and with a lesser extent digit I, played a major role in performing the end of the cycle of locomotion. Drag traces affecting the most medial digits could be formed during the recovery of the autopod at the end of the step.

Figure 12 Speculative in vivo reconstruction (based on Dinodontosaurus) of a kannemeyeriiformes dicynodont, a most probable producer of Pentasauropus tracks.

Reconstruction in back (A) and lateral (B) view of the trackmaker walking in amble gait. See the supplementary video to get a more complete view of the reconstruction. Artwork by Fabio Manucci.

A functionally plantiportal posture has been described in several mammals regardless of body-weight (e.g., South-American coati, aardvark, armadillo, coypu, among others; see Michilsens et al., 2009) but also can represent a functional strategy, co-occurring with a graviportal structure of the limbs. A subunguligrade-plantiportal foot implies a complex set of associated characters in the autopodial anatomy of the Pentasauropus producer. Body mass of taxa similar to the alleged producer of Pentasauropus have been estimated to be 23–32 kg (based on a juvenile individual of Dinodontosaurus; see Morato, 2006), 170 kg (based on an adult specimen of Dinodontosaurus brevirostris; see Mancuso et al., 2014) up to 362 kg (based on an adult individual of Dinodontosaurus platyceps; see Mancuso et al., 2014). The subunguligrade-plantiportal autopod posture was most likely promoted in these dicynodonts and in the putative producer of Pentasauropus regardless of the body-dimension. Thus, this character not necessarily implies an increase in body size but it is a pre-requisite for those lineages which experienced an increase in body-dimension.

The Vera Formation and the track record: chronostratigraphical observations

As before stated, Pentasauropus or Pentasauropus-like footprints were reported to date mainly from Upper Triassic units. In Argentina, Pentasauropus tracks were reported both from Upper Triassic unit (e.g., Portezuelo Formation) and from late Middle Triassic unit (Cerro de Las Cabras Formation). In Lesotho (Southern Africa), Pentasauropus was reported from the lower Elliot Formation (Stormberg Group), which lies above the Carnian Molteno Formation. The lower Elliot Formation was considered Upper Triassic by Ellenberger (1970), Norian-Rhaetian by Olsen & Galton (1984) and Norian by Knoll (2004), Lucas & Hancox (2001) and Lucas (2018), based on fossil remains, both bones and trace fossils. Recently, the Elliot Formation (lower and upper) was discussed by means of magnetostratigraphy, and fixed as Upper Triassic - Lower Jurassic by Sciscio et al. (2017). The same authors confirmed a Norian-Rhaetian age for the lower Elliot Formation and correlated the unit with the Los Colorados Formation in the Ischigualasto-Villa Union Basin of Argentina (Sciscio et al., 2017). A Late Triassic age of Pentasauropus-bearing levels of the Vera Formation (most probably the upper portion of this lithostratigraphic unit) is here accepted based on the currently shared distribution of the ichnogenus.

At the same time, the recent datations provided by Luppo et al. (2017) contrast with the Late Triassic age historically proposed for the whole Vera Formation and in particular for the deposits bearing the ‘Dicroidium’ flora and the ichnogenus Dicynodontipus. On the basis of the new isotopic ages, Luppo et al. (2017) concluded that at least some of the levels bearing the ‘Dicroidium’-type flora (Artabe, 1985a; Artabe, 1985b) are intercalated between deposits dated 252 ± 2 Ma (Changhsingian, Late Permian) and 248 ± 2 Ma (Olenekian, Early Triassic). These authors also suggested that the stratigraphic position of the deposits exposed in the Tchering quarry, west of Los Menucos town, where Dicynodontipus (sensu Melchor & de Valais, 2006) come from, is not yet completely clear. Nevertheless, this quarry is spatially close to the outcrops where geochronological data were provided by Luppo et al. (2017). On the other hand, the Yancaqueo farm from which the Pentasauropus footprints come, is located east of Los Menucos town and lacks detailed geochronological and geological studies.

Thus, taking into account these new data and the chronostratigraphical distribution of Dicynodontipus (e.g., Haubold, 1983; Ceoloni et al., 1988; Retallack, 1996; De Klerk, 2002; Marsicano et al., 2004; Hunt & Lucas, 2007; Klein & Lucas, 2010; Costa da Silva, Sedor & Sequeira Fernandes, 2012; Fichter & Kunz, 2013; Díaz-Martínez et al., 2015; Francischini et al., 2018), the historically proposed Late Triassic age for the the strata of the Vera Formation bearing Dicynodontipus (most likely the lower portion of the unit) is here questioned.

Conclusions

Large pentadactyl tracks from the Upper Triassic Vera Formation of the Los Menucos Group (Río Negro province, North Patagonia, Argentina) were studied and discussed in terms of palaeobiological attribution.

The tracks are currently referred to as Pentasauropus (Ellenberger, 1970), an ichnotaxon established from the Upper Triassic lower Elliot Formation (Stormberg Group) of Karoo Basin (Lesotho, Southern Africa).

Material under study allowed us to more effectively appreciate ichnotaxon variability and proved to be significant for a better definition of the locomotor dynamics of the producer and particularly of its foot anatomy.

Track and trackway parameters indicate a dicynodont as the most probable producer, and a relationship with the South-American members of the clade Kannemeyeriiformes is proposed.

An affinity between the Gondwanan therapsid ichnofauna and that from South Africa is evident, as well as functional features of the autopods of the producer are considered significantly similar and may be related to the same autopodial anatomy shared by the clade.

The autopod posture for the Pentasauropus trackmaker has been interpreted as subunguligrade in static posture and plantiportal during locomotion. A large pad of connective tissue behind the basipodials and partially metapodials can be proposed for the heavy-footed producers of Pentasauropus. The cushion allowed to decrease the stress transferred to the bones and spread it on a larger area during the touch-down and weight-bearing phase of the locomotion cycle.

Finally, a Late Triassic age for the Pentasauropus-bearing levels of the Vera Formation is accepted, based on the age of other lithostratigraphic units bearing Pentasauropus in South Africa and United States. At the same time, a detailed stratigraphic study of the lower strata of the Vera Formation, bearing-Dicynodontipus, is needed to corroborate palaeontological and geochronological data and to account the validity of the Vera Formation as lithostratigraphic unit.

Supplemental Information

Supplemental Information 1 Limb and autopod posture in the Pentasauropus trackmaker

Simplified and in vivo speculative reconstruction (based on Dinodontosaurus) showing the semi erect posture of fore and hind limbs and the subunguligrade plantiportal posture of the autopods in the Pentasauropus trackmaker. In colour the possible extension of the fleshy cushion on which the basipodials rested, ensuring support during locomotion. Artwork by Fabio Manucci.

Click here for additional data file.

I. Cerda and C. Muñóz of the Museo Provincial Carlos Ameghino (Cipolletti, Río Negro province, Argentina), D. Ramos and S. Mercado of the Museo Municipal de Los Menucos (Los Menucos, Río Negro province, Argentina) and R. Rial of the Museo Provincial María Inés Kopp (Valcheta, Río Negro province, Argentina) are kindly acknowledged for having made access to Los Menucos material possible and for their assistance during museum operations. U. Nicosia is warmly thanked for having reviewed an advanced draft of the manuscript. C.F. Kammerer is sincerely thanked for having discussed some aspects of the autopod osteology in kannemeyeriiformes. S. Lucas, H. Klein, P. Dentzien-Dias and the Academic Editor K. De Baets are acknowledged for having provided revisions that significantly improved the manuscript. F. Manucci is thanked for having performed the artworks.

Institutional abbreviations

LES Laboratoire de Paléontologie, Institut de Sciences de l’Evolution of the University of Montpellier II collection, Montpellier, France

MPCA Museo Provincial Carlos Ameghino, Cipolletti, Río Negro province, Argentina

MMLM Museo Municipal de Los Menucos, Los Menucos, Río Negro province, Argentina

MMLM (ex MRPV) Museo Provincial María Inés Kopp, Valcheta, Río Negro province, Argentina

Additional Information and Declarations

Competing Interests

Author Contributions

Data Availability

The authors declare there are no competing interests.

Paolo Citton, Ignacio Díaz-Martínez, Silvina de Valais and Carlos Cónsole-Gonella conceived and designed the experiments, performed the experiments, analyzed the data, contributed reagents/materials/analysis tools, prepared figures and/or tables, authored or reviewed drafts of the paper, approved the final draft.

The following information was supplied regarding data availability:

Citton, Paolo; Díaz-Martínez, Ignacio; de Valais, Silvina; Cónsole-Gonella, Carlos (2018): Pentadactyl tracks from Los Menucos (Patagonia Argentina). figshare. Fileset. https://doi.org/10.6084/m9.figshare.5931715.

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
