# Peer review of "Triassic pentadactyl tracks from the Los Menucos Group (Río Negro province, Patagonia Argentina): possible constraints on the autopodial posture of Gondwanan trackmakers"

_PeerJ, doi:10.7717/peerj.5358_

## Round 0.1 · original submission · Major Revisions

You provide an exhaustive study of Triassic pentadactyl tracks from Argentina which potentially reveal some interesting constraints on the posture of their producers using the latest techniques. I would like to see this work published, but there are several crucial points which need to be addressed before publication. The main points are:

Structure: The manuscript would be easier to follow and more concise if the introduction and geological setting would focus more on the trackways and the layers which contain them (see comments by reviewer 2). Placing various measurements and specifications into tables would also help in this endeavor (see below). The abbreviations in the Material and Methods should also be moved to come before the first use of them. See also further suggestions made by Reviewer 2.

Constraints on the trackmaker and its posture: I feel the tracks are well documented using the latest methods. Most reviewers agree that your interpretation could be correct. However, I miss certain metrics which demonstrate that this is not a case of overstepping and that the tracks accurately constrain the trackmaker and its posture (see comments by reviewer 1). You need to demonstrate quantitatively that Kannemeyeriiformes or related therapsids form the best candidate for these tracks and rule out other. Furthermore, you needs to more clearly demonstrate these traces accurately represent their posture or at least rule other interpretations by providing additional metrics (see comments by reviewer 1). Also a more thorough comparison of juvenile and adult tracks would be appropriate (see comments by reviewer 3).

Title: The title could potentially be shortened and could be formulated more carefully in relationship with interpretative nature of constraints on the posture. Assigning the correct producer is already not straightforward and turning it around (using traces to constrain posture of its producer) becomes even more difficult.

Measurements: Please provide measurements (e.g., Lines 359-387) as well as details of photogrammetry (lines 238-246) as tables.
Figures: I think all figures are appropriate and nicely executed. However, the discussion on the stratigraphy is hard to follow for people not familiar with it, so it would be benefit from adding a figure which shows the regional stratigraphy, age assignments and potential correlations. This could be combined with Figure 1 (this figure could be made smaller without losing information you want to convey) and would the discussion shorter as you could refer to the figure for at least some of your statements. Additionally, a figure showing the paleogeography of Pentasauropus and similar tracks would be appropriate also (see comments by reviewer 3).

Language: There are some passages which are difficult to understand. I pointed out some of them in the annotated pdf, but as I am not a native speaker myself, it would be beneficial to let a colleague with English as mother language read it before resubmission (see also comments by reviewers 2 and 3).

Please address all my comments in the annotated pdf in addition to these and all additional points raised by reviewers.

·

Basic reporting

This manuscript presents important new records of Pentasauropus that merit publication if analyzed thoroughly. However, the analysis here is far from thorough. With a more comprehensive analysis, as discussed below, this paper will become an important contribution.

Spencer Lucas

Thus, the point of the article seems to be to identify a dicynodont with subunguligrade feet with pads as the trackmaker. That may be correct, but the analysis of the footprints to demonstrate this is far from complete. Thus, looking at the illustrated tracks, those that are parts of trackways look to have short stride lengths for such large tracks, making me think that overstepping may be happening—but this is not discussed. Indeed, the dynamic potential of the trackways needs to be realized through some metrics to tell us more about the trackmaker and its posture BASED ON THE TRACKS, not based on some general ideas about dicynodont locomotion. This is a serious flaw in the manuscript as it fails to really bring the tracks to bear on its main thesis. I encourage the authors to revise the manuscript to do just that—really show how the tracks constrain the trackmaker and its posture.

I also think that many of the tracks illustrated in Figs 4-9 tell us very little, and those illustrations could be eliminated. The ones to keep are Figures 4c-d, 5a-d, 6g-h and 8a-d.

I have made some edits to the English grammar and also have these suggestions/comments keyed to numbers on the pdf:

1. So, how to resolve this conflict? Aren’t the numbers obviously wrong? (as stated by Lucas, 2018 in Tanner edited volume on the Late Triassic published by Springer).
2. These many measurements would be easier to understand in a table.
3. Also see Hunt et al. 2018 in the Tanner volume.
4. This is the problem—some statements are made here but no real analysis is presented. Are the measurements referred to here published? Please greatly amplify this discussion with documentation. Show us how the tracks from Argentina demonstrate/support your conclusions.
5. But, see Lucas 2018. The numbers are obviously wrong and should never have been published.
6. What about age assignments published by Lucas 1998, 2010?
7. This idea has been revised by Hopson 2015 in discussing pelycosaur tracks and their locomotion.
8. What does this mean? Please rewrite to clarify.
9. So, here you are advocating an elephant-like foot, so say that here. Then, what about overstepping?
10. What does this mean? Please rewrite to clarify.
11. So, the presence of sesamoids is inconclusive, or did Sidor simply overlook them?
12. Norian according to Lucas and Hancox (also see Lucas, 2018).

Experimental design

See above--the article fails because it does not bring the footprints it documents to actually bear on the problem it claims to solve.

Validity of the findings

May be valid, but need to be documented (see above)

Additional comments

Please improve this paper as suggested to make an important contribution.

·

Basic reporting

This is some important footprint material that increases our knowledge on Gondwanan tetrapod faunas.It is worth to be published in PeerJ after improvement.

Generally the author's arguments for a dicynodont trackmaker of the presented material and for the proposed anatomy and foot posture, are conclusive.

The English needs to be largely improved. The authors should contact a native English speaking person and let him make a complete re-check of the language, before resubmitting the revised manuscript.

The authors give sufficient introduction and background. Literature is fine, a few additional (optional) references I have proposed.

The article structure needs much improvement. Thereby the manuscript can be shortened widely. In my opinion it is too long. This concerns the following parts (see also attached manuscript with comments):

1) Introduction. The general statement about importance of tetrapod ichnology includes too many citations. Most of them maybe cancelled. Reduce to one or two.

2)Material and Methods. Shorten largely; some parts of the photogrammetry technique explanation maybe redundant.

3) Abbreviations. Put these right after introduction. Explanations of acronyms have to occur before they are mentioned for the first time. Then you can delete this in Material and Methods and explanations in brackets there. This part will be shortened as well.

4) Sedimentological observations. Put this part with the subhead under Geological setting.

5) Track preservation. Move the first paragraph (listing of the slabs) to Material and Methods.

6) Description. The authors give extensive measurement details for each track (line 364-387). This should be better placed in a table, which is more easy to read. Also the description part will be shortened largely. Some parts of the description should be moved to discussion (see comments in the manuscript). Some descriptions are confusing and should be re-phrased.

7) Discussion. Shorten the discussion largely.

8) Zoological attribution. Line 427-433. Trackmaker attribution by former workers can be given in one sentence. The authors do not have to mention all Pentasauropus ichnospecies again with all former interpretations.

9) A biochronological/ichnostratigraphical usefulness of Pentasauropus is questionable, because this ichnogenus is not well-established presently as a global marker. Furthermore, the authors stress that the definite ichnotaxonomic assignment of the Los Menucos tracks has to be cleared in future studies. Also Dicynodontipus is known from the Late Permian to the Late Triassic. Against the background of the uncertain age of the unit (?Late Permian-Early Triassic-Late Triassic) and after more recent dating, the authors should be careful.

10) Conclusions. Sentences are too extensive. Some statements were repeated from discussion. Authors should give main points only, in a few short sentences.

11) A stratigraphic section (general overview of formations with track levels) would be good in Fig.1 added to the map. Also give a table with measurements (see comment above).

12) Further, minor points and some language improvement are in the attached manuscript.

Experimental design

No comment.

Validity of the findings

To Conclusions see my comment above (point 10).

·

Basic reporting

-English: is good, but a native speaker must review.
- Intro & review: A good review of Los Menucos Group is made. However, the occurrence of Pentasauropus in North America is missing.
Hunt-Foster R.K., Lockley M.G., Milner A.R.C., Foster J.R., Matthews N.A., Breithaupt B.H. & Smith J.A. (2017). Tracking dinosaurs in BLM Canyon Country, Utah. Geology of the Intermountain West, 3: 67-100.
Lockley M.G. & Hunt A.P. (1995). Dinosaur tracks and other fossil footprints of the western United States. 338 pp. Columbia University Press, New York. Pages 77 and 80.
A paper published last year from the type locality of Pentasauropus of the Elliot Formation (Lesotho) must be cited.
Bordy et al., 2017. The Subeng vertebrate tracks: stratigraphy, sedimentology and a digital archive of a historic Upper Triassic palaeosurface (lower Elliot Formation), Leribe, Lesotho (southern Africa). Bollettino della Società Paleontologica Italiana, 56 (2), 181-198.
Pentasauropus-like tracks occur in Poland. See Klein H. & Niedzwiedzki G. (2012). Revision of the Lower Triassic tetrapod ichnofauna from Wiory, Holy Cross Mountains, Poland. New Mexico Museum of Natural History and Science Bulletin, 56: 1-62. Figure 52E.
- Structure: A table with the measurements is needed to make it easy to interpreter the data. Improvements in the captions and figures should be made (see comments in the PDF).
- Figures: Figures are very good and have a good quality. However, a paleogeographic map showing the distribution of Pentasauropus is necessary (including the new references listed above), and a scatch comparing the juvenile with the adult tracks would greatly improve the manuscript. The different morphologies from each slab should be indicated in the caption and/or in the image.
The captions can be improved, see more comments in the PDF.

Experimental design

The manuscript is original, with a large number of footprints never studied before and in the scope of PeerJ.

Validity of the findings

- The research is relevant and improves the knowledge of dicinodonts and made a good description of 60 footprints classified as Pentasauropus from Argentina (which was only briefly reported). The studied material allowed the recognition of ichnotaxon variability, involving the differences of juvenile and adult, locomotor dynamics and foot anatomy.
- The techniques used are very good, with three-dimensional representations and interpretative drawings of the studied material.
- Methods are well described.

Additional comments

See PDF for more comments.

---

## Round 0.2 · Minor Revisions

Thank you for addressing our suggestions. Particularly, the added measurements and restructuring which have made the manuscript easier to follow. There are still some minor points I would like to see addressed before publication:

1) Jargon/Terminology: you use quite some specific terms from ichnology and to a lesser extent anatomy which not all readers might be able to follow. The use of these terms is of course normal. However, as this manuscript is of interest to a broad audience including zoologists, paleobiologists and ichnologists, I would like to see you briefly define them they first time these are used. This could be done be “briefly” defining the term or more add a more generally synonym the first time they are used (at least for the most important and commonly used ones; see annotated pdf for some examples). You can take a look at Francischini et al. (2018) for examples how it was done here.

2) Possibility of overstepping: it was a necessary step to mention the possibility of overstepping. Your interpretation that overstepping might not be issue might be justified but it is currently hard to follow for the readers who are not familiar with it. I would like you to mention more specifically in which cases (specimen numbers, particular tracks) overstepping might be issue. Furthermore, it would be good to mention more explicitly why you think overstepping is unlikely in your case and how it could influence your interpretation of the autopodial posture of the track makers (if at all).

3) Age of the Pentasauropus tracks and the Vera Formation as a whole: It would be good to mention that the Pentasauropus tracks have been attributed to the Vera Formation and also add their possible stratigraphic range on Figure 1 (if uncertain a stippled line would be ok). It would be good to use the same color for the formations on the map (1a) in Fig. 1b. There seems to be quite some lithological variation in the Vera Formation as shown in Fig. 1b. Is it really not possible to make an educated on the probably origin of your traces to a particular part of the Vera Formation based on lithology. Also, do the designated ignimbrites and tuffites in Fig. 1b refer to any of the dated material? It would be good to add those ages too. As for the difference between the layers containing Pentasauropus and Dicynodontipus, it would also be good to mention Francischini et al. (2018) who mention the presence of Dicynodontipus in the Permian-Triassic (similar to the age range of Luppo et al. (2017). Would it be possible that Pentasauropus comes from higher (younger) in the section than the Dicynodontipus tracks?

Further suggestions can be found in the annotated pdf. Please address these points and other suggestions in the annotated pdf.

Suggested reference:

Francischini H, Dentzien-Dias P, Lucas SG, Schultz CL. (2018) Tetrapod tracks in Permo–Triassic eolian beds of southern Brazil (Paraná Basin) PeerJ 6:e4764 https://doi.org/10.7717/peerj.4764

---

## Round 0.3 · accepted · Accept

Thank you for clarifying some of the terminology, adding the possible stratigraphic position of the tracks in figure 1 as well as discussing the possibility of overstepping in greater detail. Your manuscript is now even more easier to follow. Looking forward to seeing it published.

#